# Dendritic Cells Pulsed with HAM/TSP Exosomes Sensitize CD4 T Cells to Enhance HTLV-1 Infection, Induce Helper T-Cell Polarization, and Decrease Cytotoxic T-Cell Response

**DOI:** 10.3390/v16091443

**Published:** 2024-09-10

**Authors:** Julie Joseph, Thomas A. Premeaux, Ritesh Tandon, Edward L. Murphy, Roberta Bruhn, Christophe Nicot, Bobby Brooke Herrera, Alexander Lemenze, Reem Alatrash, Prince Baffour Tonto, Lishomwa C. Ndhlovu, Pooja Jain

**Affiliations:** 1Department of Microbiology & Immunology, Drexel University College of Medicine, Philadelphia, PA 19129, USA; 2Department of Medicine, Division of Infectious Diseases, Weill Cornel Medicine, New York, NY 10021, USA; tap4002@med.cornell.edu (T.A.P.); lndhlovu@med.cornell.edu (L.C.N.); 3Departments of Laboratory Medicine and Epidemiology/Biostatistics, University of California, San Francisco, CA 94143, USA; 4Vitalant Research Institute, San Francisco, CA 94105, USA; 5Department of Pathology and Laboratory Medicine, University of Kansas Medical Center, Kansas City, KS 66103, USA; cnicot@kumc.edu; 6Rutgers Global Health Institute, Rutgers University, Newark, NJ 07102, USApb730@rwjms.rutgers.edu (P.B.T.); 7Department of Medicine, Division of Allergy, Immunology, and Infectious Diseases, and Child Health Institute of New Jersey, Rutgers Robert Wood Johnson Medical School, Rutgers University, New Brunswick, NJ 08901, USA; 8Molecular and Genomics Informatics Core, Rutgers New Jersey Medical School, Newark, NJ 07103, USA; 9Department of Neurobiology and Anatomy, Drexel University College of Medicine, Philadelphia, PA 19129, USA

**Keywords:** HAM/TSP, exosomes, HTLV-1, dendritic cells, T cells

## Abstract

HTLV-1-associated myelopathy/tropical spastic paraparesis (HAM/TSP) is a progressive demyelinating disease of the spinal cord due to chronic inflammation. Hallmarks of disease pathology include dysfunctional anti-viral responses and the infiltration of HTLV-1-infected CD4+ T cells and HTLV-1-specific CD8+ T cells in the central nervous system. HAM/TSP individuals exhibit CD4+ and CD8+ T cells with elevated co-expression of multiple inhibitory immune checkpoint proteins (ICPs), but ICP blockade strategies can only partially restore CD8+ T-cell effector function. Exosomes, small extracellular vesicles, can enhance the spread of viral infections and blunt anti-viral responses. Here, we evaluated the impact of exosomes isolated from HTLV-1-infected cells and HAM/TSP patient sera on dendritic cell (DC) and T-cell phenotypes and function. We observed that exosomes derived from HTLV-infected cell lines (OSP2) elicit proinflammatory cytokine responses in DCs, promote helper CD4+ T-cell polarization, and suppress CD8+ T-cell effector function. Furthermore, exosomes from individuals with HAM/TSP stimulate CD4+ T-cell polarization, marked by increased Th1 and regulatory T-cell differentiation. We conclude that exosomes in the setting of HAM/TSP are detrimental to DC and T-cell function and may contribute to the progression of pathology with HTLV-1 infection.

## 1. Introduction

Human T-lymphotropic virus type 1 (HTLV-1)-associated myelopathy/tropical spastic paraparesis (HAM/TSP) is a debilitating neurodegenerative viral-immune-mediated disorder of the spinal cord. An estimated 0.2–5% of people infected with HTLV-1 will develop HAM/TSP [1]. Disease progression is due to chronic inflammation and infiltration of HTLV-1-infected CD4+ T cells and HTLV-1-specific CD8+ T cells in the central nervous system (CNS). A prominent feature of HAM/TSP development is elevated levels of HTLV-specific cytotoxic T lymphocytes (CTLs) in the blood compared to asymptomatic carriers (ACs) [2] and increased expression of inhibitory immune checkpoint proteins (ICPs) associated with cellular exhaustion [3,4,5,6]. CD8+ T cells specific to Tax, the immunodominant HTLV antigen, are at a higher frequency within the cerebrospinal fluid (CSF) than the periphery [7,8]. During the early stages of HAM/TSP, CD4+ T cells, high proviral load (PVL), and elevated Tax and IFN-γ expression dominate within the CNS. The CD4+ T-cell anti-viral response in HAM/TSP is Th1-dominant [9], which can initiate a proinflammatory feedback loop in stimulating astrocytes within the CNS to recruit and promote T-cell infiltration, further perpetuating tissue damage. HTLV-1 preferentially infects CD25-expressing CD4+ T cells, often considered regulatory T cells (Tregs), which contribute to self-tolerance and dampen hyperactive immune responses. CD4+CD25+ T cells in HAM/TSP patients carry a high PVL and stimulate HTLV-specific CTLs; however, their expression of FoxP3 is reduced, impairing their immunosuppressive and regulatory function.

Immune cell-derived exosomes, including those derived from infected cells, can modulate innate and adaptive responses [10,11]. Exosomes can carry inhibitory ICPs, with functional implications on the cellular microenvironment and immune system [12,13]. In metastatic melanoma, there is a high release of exosomes carrying PD-L1 on their surface, resulting in increased tumor growth and tumor-specific CD8+ T cell suppression [14,15]. In the nervous system, exosomes can participate in intercellular communication, maintain the myelin sheath, and eliminate waste, as well as drive positive or negative effects in numerous CNS complications [16,17,18,19,20]. T-lymphocyte-derived exosomes induce TCR stimulation and apoptosis and regulate the functions of other immune cells in an antigen-specific manner [21]. Immune cell-derived exosomes act as a double-edged sword by enhancing cell persistence and increasing proliferation in autologous cells while also influencing inflammatory responses, anti-tumor responses, and drug resistance [22]. However, little is known regarding the role of exosomes in the pathogenesis of HAM/TSP and their influence on dendritic cell (DC) and T-cell responses.

DCs are key mediators of priming and maintaining T-cell-mediated responses. Studies indicate that exosomes derived from tumors have the potential to suppress anti-cancer responses, as well as undermine DC differentiation, maturation, and functional responses [23]. In the context of HTLV-1, DC subsets, including plasmacytoid DCs (pDCs), myeloid DCs (mDCs), and monocyte-derived DCs (moDCs), are permissible in HTLV-1 infections [24,25,26]. DCs can further propagate infection as they travel to the lymph node, and DCs isolated from HTLV-1-positive patients are defective in alpha interferon (IFN-α) response upon in vitro stimulation, while their ability to activate T cells is also impaired [26,27]. In contrast to mDCs, pDCs are impaired in transmitting HTLV-1 to CD4+ T cells, suggesting that they may contribute to a protective immune response [24]. pDCs bypass viral escape due to their type 1 interferon responses and their activation inversely correlates with the ability of infected cells to transmit the virus. Interestingly, DC-mediated viral transfer often results in sustained infection and IL-2-independent transformation of CD4+ T cells [27]. Evidence supports the role of DCs in stimulating autologous T-cell activity in HAM/TSP [28]. Furthermore, the expression of multiple negative checkpoint receptor/ligands (NCR) on mDCs and pDCs in clinical samples may impact their role in initiating anti-viral responses [2]. Interestingly, DC-derived exosomes can exhibit both immune-stimulatory and -suppressive effects and have become increasingly relevant in cell-free therapeutics [29,30]. Exosome-pulsed DCs can stimulate antigen-specific CD8+ T-cell proliferation in vitro and in vivo [31]. DC access to CNS antigens can play a significant role in immune-mediated disorders of the CNS [32]. However, its relevance to HAM/TSP-linked neuroinflammation past enhancing spread remains to be understood. Determining the role that exosomes derived from HTLV-infected cells, which can carry both viral particles and immunosuppressive mediators, play in regulating immune function can potentially elucidate underlying mechanisms involved in disease progression. In this study, we aimed to determine the effects of HAM/TSP-derived exosomes from cell lines and patient samples on DC function, helper CD4+ T-cell polarization, and CD8+ T-cell cytotoxic activity.

## 2. Materials and Methods

### 2.1. Study Participant Details

A subset of participants from the University of California San Francisco (UCSF) HTLV Outcomes Study (HOST) cohort were utilized for ex vivo analysis. Details of the HOST cohort enrollment have been previously described [33]. Briefly, HOST is a longitudinal prospective study of HTLV-1 and -II seropositive blood donors and seronegative controls identified from 1988 to 1992 and followed prospectively (detailed description of the cohort study reviewed in [34]) All participants included in this study had detectable HTLV-1 at the time of blood donation. Participants with HAM/TSP were matched to asymptomatic HTLV-1 seropositive individuals for age, sex, and race.

### 2.2. Cell Lines

The OSP2 (HAM/TSP-derived) cell cultures were acquired through the NIH AIDS Reagent Program (Cat. No. ARP-1393). All cell lines were cultured in complete media consisting of RPMI (Gibco, Frederick, MD, USA) with 15% heat inactivated FBS (Biotechne, Minneapolis, MN, USA), supplemented with 1% Penicillin Streptomycin (Gibco), and 10 mM HEPES (Gibco). OPS2 cells were additionally supplemented with IL-2 (Sigma, St. Louis, MO, USA) at 10 units/mL. All cultures were routinely tested for mycoplasma (Applied Biological Materials, Richmond, BC, Canada).

### 2.3. Exosome Preparation and Quantification

Cells were seeded at densities of 3–5 × 10^5^/mL in complete media with 10% exosome free-FBS (Thermofisher, Waltham, MA, USA), and supernatants were collected on day 3. Differential centrifugation at 400× *g* and 2000× *g* for 10 min was performed to remove cells and debris. The resulting supernatant was passed through a 0.45-micron filter and directly centrifuged at 100,000× *g* for 2 h. EV pellets were resuspended, washed in 2 mL DPBS, and ultracentrifuged at 27,000 RPM (100,000× *g*) for 75 min at 4 °C. Isolated exosomes were resuspended in DPBS at a desired volume and subjected to protein quantification by the BCA protein assay (Thermofisher, Waltham, MA, USA). Characterization of exosomes derived from viral protein knocked down lines was performed as previously described [35].

Serum samples (2 mL) from patients were consecutively centrifuged at 400× *g*, 2000× *g*, and 10,000× *g* at 4 °C to remove debris and fat content. The samples were then passed through a 0.45-micron membrane filter before exosome collection via AFC Izon (Izon Science Limited© 2023, Christchurch, New Zealand) column filtration. Briefly, the AFC system was calibrated per manufacturers instruction. Collection volume was set to 0.8 mL per fraction after 3.0 mL void volume using qEV (70 nm) columns. The fractions were then analyzed by Tunable Resistive Pulse Sensing (TRPS) (Izon Science Limited© 2023, Christchurch, New Zealand) and ZetaView X30 TWIN system© ParticleMatrix. EVs particle sizes were determined using the ZetaViewX30 TWIN by Particle Matrix: For each measurement, the instrument pre-acquisition parameters were set to a temperature of 25 °C, 488 nm laser wavelength, a sensitivity of 85, 30 frames per second, and a shutter speed of 70. For each measurement, 1 mL of diluted sample (1:200) in deionized (DI) water was loaded into the cell, and the instrument measured each sample at 11 different positions throughout the cell, with three readings at each position and two distribution cycles. After automated analysis and removal of any outliers from the 11 positions were completed, the concentration, mean, median, and mode sizes of the samples were calculated using the ZetaView 8.04.02 software and analyzed using the same software and Microsoft Excel v16.61.1. As published before [35], every exosome preparation was tested to confirm that HTLV-1 virions were not precipitated using a specific anti-p19 ELISA. The HTLV-1 matrix protein (p19) often associated with virus production was not detected in the exosome isolations from HTLV-1-infected cell lines, confirming that virus was not concentrated in high quantities during the isolation process [35].

### 2.4. Cell Isolation

Utilizing the Pan-DC Enrichment Kit-human (Miltenyi Biotec, Bergisch Gladbach, Germany, Cat. 130-100-777), DCs (MDC1s, MDC2s, and PDCs) were isolated from the PBMCs of normal donors using the buffy coat (500 μL). Briefly, non-target cells were labeled with a pre-made cocktail (100 μL per 10^7^ million cells) of biotin-conjugated monoclonal antibodies and anti-biotin and magnetic labeled microbeads against antigens that are not expressed by DCs. DCs were negatively selected for when magnetically labeled non-target cells were depleted by magnetic separation. Similarly, EasySep™ Human CD3 Positive Selection Kit II (Cat. 100-0692) was used per manufacturer’s instructions for T-cell isolations. The purify of isolated T cells and DC ranged from 80 to 100%. A ratio of 1:50 cells was maintained for the DC–T-cell experiments.

### 2.5. Stimulation Assays

Purified exosomes from OSP2 cells (without or with inhibitor treatment, Manumycin A (BioViotica, Liestal, Switzerland; 250 nM in DMSO) or from patient sera were mixed (1:1 (~20 μg)) with positively selected (StemCell, Seattle, WA, USA) CD3+ T cells activated with a CD3/CD28 activation cocktail (1×, BioLegend, San Diego, CA, USA) from healthy donors (HumanCellsBio, Milpitas, CA, USA) for 24 h. Where indicated, exosomes were pre-incubated with an ICP blockade at 1:100 dilution (BTLA (Cat. 7473, ProSci, Poway, CA, USA); PD-L2 (Cat. 4063, ProSci); LAG-3 (Cat. 13485, Raybiotech, Peachtree Corners, GA, USA)). CD3+ T cells and total DCs were isolated from matched donors (*n* = 4), as described above. The DCs were then stimulated with 20 μg of exosomes from HTLV-1-infected OSP2 cells or patient sera for 48 h and then co-incubated with donor-matched T cells for another 24 h.

### 2.6. Flow Cytometry

For phenotyping, cells were collected, washed with PBS, blocked for 20 min with Fc block (BioLegend, San Diego, CA, USA), and then stained with fluorochrome-conjugated anti-human antibodies for various surface markers. Subsequently, for intracellular staining, cells were washed, fixed and permeabilized (Invitrogen, Carlsbad, CA, USA) for 20 min, and then incubated with fluorochrome-conjugated antibodies to specific targets of interest in a 1× permeabilization buffer. Details on the DC and T-cell panels, including the antibodies and corresponding channels, are listed in Appendix A, respectively. Data were acquired on a Fortessa LSR (BD) and analyzed using FlowJo software (v10.9).

### 2.7. Real-Time Quantitative PCR

Total RNA was isolated from cells using the TRIzol Plus RNA Purification Kit (Invitrogen™, Carlsbad, CA, USA). Briefly, 1 mL of TRIzol reagent per million cells was added to the cell pellet and incubated for 5 min at room temperature. Chloroform was added (0.2 mL per 1 mL of TRIzol), incubated for 2–3 min, and then centrifuged for 15 min at 12,000× *g* at 4 °C. The upper aqueous phase containing RNA was transferred to a new tube with an equal volume of 70% ethanol, vortexed, and subsequently transferred to a spin column followed by centrifugal and wash cycles per manufacturer’s instructions. RNA was eluted with RNase-free water, and concentration was determined via Nanodrop. Reverse transcription PCR was performed using complementary DNA (cDNA) with a High-Capacity cDNA Reverse Transcription Kit (Applied Biosystems, Waltham, MA, USA). Then, 6 ng of cDNA was loaded onto a TaqMan™ Array Human Cytokine Network (Applied Biosystems Cat. 4414124) per target. Real-time PCR was performed using the Applied Biosystems QuantStudio 3 Real-Time PCR machine.

### 2.8. Immunoassays

Biomarkers in the supernatants from the stimulated mDC and pDC cultures were quantified using a custom bead-based multiplex immunoassay according to the manufacturer’s instructions (ThermoFisher, Cat. PPX-12, Waltham, MA, USA) and included the following antibodies: interferon alpha (IFN-α), IFN-β, interleukin-1 beta (IL-1β), IL-4, IL-6, IL-10, IL-12p70, IL-13, IL-21, IL-23, IL-27, and tumor necrosis factor alpha (TNF-α). Data were acquired on a Luminex 200^TM^ analyzer and analyzed using the MILLIPLEX^®^ Analyst software v5.1 (Millipore, Burlington, MA, USA).

To quantify extracellular proinflammatory cytokine levels of cell line supernatants and exosomes, enzyme-linked immunosorbent assays (ELISAs) were performed in triplicate, measuring the levels of IFN-γ (ELISA Max Deluxe Set, BioLegend, Cat. 430104), TNF-α (Invitrogen, Cat. BMS223-4), and transforming growth factor beta (TGF-β; Invitrogen, Cat# BMS249-4), PD-1, PD-L2, BTLA (Human ELISA kit, Invitrogen Cat. BMS2214, BMS2215, BMS2217), and LAG-3 (Human LAG-3 ELISA kit, RayBio Cat. ELH-LAG3-1). Absorbance measurements and standard curve extrapolation were conducted on Tecan Magellan software (Männedorf, Switzerland).

### 2.9. Immunoblotting

Whole-cell lysates were extracted via RIPA lysis (ChemCruz, Dallas, TX, USA) and quantified for the protein yield. A total of 10 µg of whole cell lysate or 15–25 µg of exosomal protein was loaded onto SDS-PAGE gels (Bio-Rad, Hercules, CA, USA); transferred onto methanol-activated PVDF membranes; blocked in 5% non-fat milk in Tris-buffered saline (TBS); washed in 1% TBS-Tween; and incubated with primary antibody (p-Akt (Proteintech, Rosemont, IL, USA, Cat. 66444-1-Ig), p-Erk (Proteintech, Cat. 24390-1-AP, Rosemont, IL, USA), and beta actin (Proteintech, Cat. 66009-1-Ig, Rosemont, IL, USA)) overnight in 4 °C. After incubation, the membranes were washed and incubated with a secondary antibody conjugated to HRP for one hour. The membranes were developed with the chemiluminescent substrate (West Pico PLUS, SuperSignal Pierce) and imaged by ImageQuant LAS 4000 (Piscataway, NJ, USA).

### 2.10. Honeycomb HIVE CLX

Single-cell RNA sequencing (scRNAseq) was carried out using the HIVE CLX system from Honeycomb Bio (Waltham, MA, USA). Briefly, PBMCs were thawed down and resuspended in PBS containing 1% Bovine Serum Albumin, followed by cell counting using a Countess 3 automated cell counter (Invitrogen). A total of 250,000 cells were split into 2 equal batches and seeded into a round bottom 96-well plate. One batch was treated with 30 µg of exosomes, while the other batch received no treatment, followed by incubation for 18 h at 37 °C. Following incubation, 20,000 cells from both batches were loaded into HIVE collectors following the manufacturer’s instructions (Honeycomb Bio, Waltham, MA, USA). The HIVE collectors, containing the loaded cells, were stored at −80 °C until scRNAseq analysis.

### 2.11. Statistical Analysis

All analyses were performed in Prism v9.4 (GraphPad Software, LLC, Boston, MA, USA) and Microsoft Excel. Differences between paired or unpaired groups were analyzed using T-tests. One-way ANOVA was performed to compare differences between more than 2 independent groups. Plots were generated using GraphPad Prism, with *p* < 0.05 *, *p* < 0.005 **, and *p* < 0.001 *** as the threshold for statistical significance.

## 3. Results

### 3.1. Exosomes Derived from HTLV-Infected Cell Lines Elicit Proinflammatory Cytokine Responses in Dendritic Cells

HTLV-1 viral proteins and its immunomodulatory properties have been extensively studied [36,37]. In addition, exosomes derived from HTLV-1 infected cells can also display the viral protein Tax, which can elicit cytokine responses, particularly those that induce T cell polarization [11]. In efforts to further assess the immune-stimulating properties of HTLV-1-associated exosomes, we first evaluated their ability to induce the activation of DCs and cytokine profiles known to influence T-cell polarization trajectories. We exposed DCs isolated from HTLV-1 seronegative individuals with exosomes derived from OSP2 cells for 48 h and assessed the differences in surface expression of activation markers CD40, CD80, and CD86 (Appendix A). LPS stimulation was conducted as a positive control for DC activation (Appendix A). A significant increase in the expression of activation markers CD80 (*p* = 0.012) and CD86 (*p* = 0.002) was observed in the mDCs following exosome exposure (Figure 1A). Furthermore, the expression of CD40 (*p* = 0.004), CD80 (*p* = 0.020), and CD86 (*p* = 0.009) was significantly increased in pDCs stimulated with OSP2 exosomes. While we demonstrated that exosome uptake activates DC populations, only a slight reduction in activation markers was observed when the exosomes were pretreated with monoclonal antibodies that block ICPs.

At the transcriptional level, OSP2 exosome-exposed DCs demonstrated elevations in IFN-γ, IL-2, IL-4, IL-6, IL-17, IL-10, IL-13, and TGF-β (Figure 1B), indicating that an overall profile that can skew T cells towards a proinflammatory state. Conversely, a decrease in type 1 interferon responses with exosome stimulation was observed (Appendix A). We also observed elevated levels of IL-6, IL-13, TNF-α, and TGF-β in the supernatants of mDCs and pDCs stimulated with OSP2-derived exosomes (Figure 1C). There was a reduction in these cytokine levels when the DCs were treated with exosomes from OSP2 cells with the knockout of HBZ and Tax proteins. The HTLV-1 matrix protein (p19) often associated with virus production was not detected in the exosome isolations from the OSP-2 cell lines, as previously reported [35].

### 3.2. Exosomes Derived from HTLV-Infected Cell Lines Promote T-Cell Polarization

While exosome-mediated immune regulation is commonly observed in CTL responses, few studies have investigated the impact on CD4+ T-helper cells. Given the impact of exosomes on DC maturation and cytokine release, we then evaluated their effects on DC-mediated T-cell polarization. Total DCs pretreated with OSP2-derived exosomes were subsequently co-incubated with donor-matched CD3+ T cells. CD4+ T cells were analyzed for specific helper T-cell associated markers (Figure 2A): Th1 (IFN-γ, CCR5), Th2 (IL-4, CCR4), Th17 (IL-17a, CCR6), and Treg (FoxP3, CD25). The induction of Th2 (*p* = 0.02) and Treg (*p* = 0.001) polarization was significantly elevated following exosome exposure, as well as notably increased in the Th1 population, indicating that OSP2-derived exosomes induce T-cell activation overall. A decrease in these cell populations was observed when the exosomes were pretreated with a cocktail of mAb blocking ICPs; however, no significant changes were observed. Consistent with the T-cell polarization results, we observed an elevation in CD4+ T cells expressing IFN-γ (*p* = 0.03), IL-4 (*p* = 0.04), and CCR6 (*p* = 0.04; Figure 2B). The expression of CCR5, a chemokine also involved in the recruitment of Treg cells to sites of infection and inflammation, was significantly increased (*p* = 0.005), of importance as CCR5 activity can negatively influence Th1 and Treg responses [38]. We also observed significant increases in IFN-γ, TGF-β, and TNF-α in the supernatant, consistent with increases in Th1, Th2 and Treg cell polarization (Figure 2C).

### 3.3. Exosomes Derived from Individuals with HTLV-1 and HAM/TSP Impact T-Cell Function

We next evaluated the impact of exosomes derived from people infected with HTLV on T-cell function, specifically the expression of inhibitory ICPs and anti-viral cytokines. Demographic information on the participants is detailed in Appendix A. Due to limitations in patient sample quantity, a column-based isolation strategy was utilized (AFC, Izon) to isolate and quantify exosomes (Figure 3A). Using the Revvity and Honeycomb Bio’s single-cell genomics platform, we captured populations of immune cells from patient’ PBMCs and evaluated gene expression patterns following exposure to exosomes. We noted distinct changes in the regulation of genes associated with immune activity [39], adhesion [40], and lipid profiles [41], which are often altered in adult T-cell leukemia/lymphoma (ATLL) and HAM/TSP patients (Appendix A).

The sera exosomes isolated from HTLV-1-infected individuals diagnosed with HAM/TSP (*n* = 3) showed a significant elevation in LAG-3 (Lymphocyte-activation gene 3) compared to AC (*n* = 1) sera-derived exosomes (*p* = 0.017; Figure 3B). Based on our previous immune checkpoint findings [35], we also evaluated BTLA (B- and T-lymphocyte attenuator), PD-1 (Programmed death-1), and PD-L1 (Programmed death ligand-1). Our results showed that these markers tended to be higher in TSP/HAM patients than AC. No significant differences in ICPs were observed among individuals infected with HTLV-2. As such, postulating whether this difference is due to the lack of specific viral proteins compared to HTLV-1, differences due to virus homology (~70% to HTLV-1), or differences in infectivity remains indefinable [42]. Individuals with HAM/TSP also had elevated IFN-γ and TNF-α levels in their exosomes compared to AC (Figure 3C).

### 3.4. Patient-Derived Exosomes Stimulate CD4+ T-Cell Polarization

We next evaluated the impact of patient-derived exosomes on CD4+ T cells using our in vitro model of OSP2 cell-derived exosomes. Isolated DCs were exposed to a pool of exosomes from people with HAM/TSP (*n* = 4) before co-culturing with donor-matched CD3+ T cells (Figure 4A, Appendix A). We observed a significant increase in Th1 (*p* = 0.024), Th2 (*p* = 0.026), and Treg (*p* = 0.030) polarization in response to exosome exposure. Additionally, the IFN-γ, TNF-α, and TGF-β levels were comparable to those of OSP2 cell-derived exosomes (Figure 4B). Similarly, in evaluating the functional differences seen in CD4+ T cells, a significant increase in IFN-γ (*p* = 0.005), IL-4 (*p* = 0.0008), CD25 (*p* = 0.040), and CCR5 (*p* = 0.010) was observed (Figure 4C).

### 3.5. HAM/TSP Patients Show Increased Th1 and Treg Differentiation after Exosome Stimulation

Studying the effects of exosomes on healthy cells can reveal their role in modulating immune responses and potential in exacerbating HAM/TSP pathology. In HAM/TSP patients, the viral protein Tax manipulates CD4+ T-cell plasticity, induces transcriptional changes that suppress T-reg activity, and promotes Th1-associated inflammation [43]. Considering that exosomes also carry Tax, evaluating exosomal influence in patients can provide insights into their impact in T-cell dysfunction. Hence, we co-cultured participant-matched sera-derived exosomes and CD3+ T cells (*n* = 4) to determine the changes in Th1 (IFN-γ and CCR5), Th2 (IL-4 and CCR4), Th17 (IL-17a and CCR6), and Treg (FoxP3 and CD25) phenotypes. We observed significant increase in Th1 (*p* = 0.023) and Treg (*p* = 0.018) expression (Figure 5A), as well as increased levels of IFN-γ (*p* = 0.005), CD25 (*p* = 0.006), and CCR5 (*p* = 0.038) following exosome exposure (Figure 5B). Given the dysregulation of these markers in HAM/TSP, it is surprising that their expression profiles were similar to those of healthy individuals. However, this may be due to the differences in low proviral load of sampled PBMCs, which is known to directly correlate with T-cell dysfunction [44].

### 3.6. OSP2-Derived Exosomes Suppress CD8+ T-Cell Function

Building on our previous work revealing the correlates of inhibitory ICP expression and HTLV-specific CD8+ T-cell function [2], we next evaluated the impact of OSP2-derived exosomes on CD8+ T-cell responses. The activated CD8+ T cells from healthy donors incubated with exosomes demonstrated decreases in MIP-1α+ and TNF-α+ CD8+ T cells [35]. Comparatively, Jurkat-derived exosomes exhibited no effect on T-cell activation, indicating that these changes are specific to HTLV. Furthermore, the pretreatment of OSP2 cells with Manumycin A (an exosome inhibitor) before exosome purification did not result in a similar suppressive effect as those seen with exosomes isolated from untreated cells. These data suggest a distinct dampening of activated T cells following exosome exposure and a distinct role for exosomal ICPs in contributing to adverse T-cell function.

DCs play a central role in regulating the balance between CD8+-mediated immunity and tolerance. Further, DCs stimulated with exosomes can cross-prime CD8+ T cells and influence the efficacy of immune responses [45]. Therefore, we next evaluated whether DCs pretreated with exosomes could impact CD8+ T-cell responses. No significant changes were seen with DC-mediated CD8+ T-cell function (Figure 6A), which could be due to a threshold of antigenic burden in exosomes to cross prime CD+8 T cells or technical challenges of reduced CD8+ T-cell counts from donors. Interestingly, when direct exposure was performed after anti-CD3/CD28-based activation, a significant reduction in MIP-1α (*p* = 0.03) and Granzyme B (*p* = 0.002) was observed, along with decreased trends in Perforin and IFN-γ. However, this exosome-mediated effect was restored when T cells were pretreated with a cocktail of ICP-blocking antibodies (Figure 6B). In patient CD8+ T cells, an increase in cells expressing MIP-1α (*p* = 0.01), Granzyme B (*p* = 0.018), and IFN-γ (*p* = 0.003) was observed and was further enhanced in exosomes pretreated with the cocktail of mAb-blocking ICPs (Figure 6C).

Exosomes play significant roles in influencing multiple pathways. While numerous studies have emphasized the role of exosomal PD-1 [46] and other immune checkpoint molecules in regulating immune cell function, the mechanisms by which these molecules influence activated cells remain unclear. Exosomal PD-1/PD-L1 can functionally bind to their cognate receptor/ligand on recipient cells and induce downstream activation of AKT/ERK [46]. We demonstrated that, overall, exosomes reduce activity p-AKT and p-ERK in anti-CD3/CD28 pre-activated cells, indicating an immunosuppressive role (Figure 6D). While exosomes purified from Jurkat cells showed a reduction, it was significantly more pronounced in the presence of HTLV-infected OSP2 cell-derived exosomes. Importantly OSP2 cells produce elevated levels of several ICPs, and our studies showed that p-Akt and p-ERK activity was restored upon blockade of these ICP.

## 4. Discussion

The pathogenesis of HAM/TSP remains complex and unclear. Given that exosomes, particularly those derived from infected cells or tumors, alter the differentiation, maturation, and function of DCs and impart immunosuppressive effects, they may, in part, contribute to HTLV-1 pathological outcomes [23]. DCs bridge HTLV-1 transmission by internalizing the virus and migrating to lymphoid tissue, where they can engage the adaptive immune response and propagate the virus. Although cell-free HTLV-1 infects DCs and leads to the spread of HTLV-1 and the transformation of CD4+ T cells [27], mature DCs restrict HTLV-1 infection and transmission [25]. However, questions remain on the direct impact of EVs on HTLV-1 infection and transmission. Our novel findings add new immunological insight into the biology of EVs in HAM/TSP pathogenesis.

Physiologically, pDCs uptake exosomes more frequently than mDC, and in healthy steady-state conditions, they produce significant type I IFNs before differentiating into antigen-presenting cells. However, in chronic infections, pDCs contribute to the induction of tolerance while also facilitating an inflammatory environment [47]. Our data demonstrated that mDCs and pDCs in response to exosome stimulation release cytokines associated with inflammation and T-helper cell polarization, including IL-6 and IL-13. The secretion of IL-6, a contributor to T-cell differentiation and potent chemoattractant, with exosome stimulation that we observed could potentially be linked with the growth and maturation of DCs [48]. IL-13 secretion is often associated with Th2-type cytokine skewing, and Tax can drive IL-13 expression [49]. Exosomes can also regulate type 1 IFN responses and anti-viral activity [50]. Type 1 IFNs are important mediators of anti-viral immunity and autoimmune diseases. Although some studies have explored innate immune suppression by exosomes, particularly in viral infection, conclusions remain contradictory as heterogeneity in miRNA cargo from immune cell-derived exosomes can significantly impact TLR activation and type 1 IFN induction [22]. Additionally, our previous data demonstrated that IL-4 and TGF-β expression can impact DC susceptibility to HTLV-1 infection and type 1 IFN responses [51]. Furthermore, IFN-α-expressing DC have been shown to restrict productive infection without inhibiting virion uptake [25]. It is possible that the viral cargo within the exosome is influencing this outcome. Given the sensitivity of DCs towards HTLV-1 uptake and spread while also restricting transmission, in chronic infections, mDCs and pDCs become activated and accumulate in lymphoid tissue [52], and our data suggest that exosomes may be a contributive factor indicative of increased maturation marker expression.

Due to the ability of DCs to cross-present in both CD4+ and CD8+ T cells, their potential in priming antigen-specific T cells to enhance anti-tumor and anti-viral immunity are highly investigated [45]. HTLV-1 progression to HAM/TSP biases towards a Th1-like state. Given the role DCs have in propagating infection, exploring whether exosomes can also influence T-cell differentiation and aid in HAM/TSP progression could provide pathological insights. Our data showed that DCs pulsed with HAM/TSP exosomes initiate the differentiation of Th1 and Treg cells in healthy and infected patients while also producing proinflammatory cytokines. Unsurprisingly, the dysregulation of CD4+CD25+CCR4+ T cells was observed among individuals with HAM/TSP [43]. However, a prolonged exposure analysis would be necessary to delineate which subset of populations are specifically enhanced with time compared to isolated exposure in in vitro systems. Several studies have eluded to the induction of Tregs via exosome stimulation promoting immunosuppressive properties [53]. Conversely, certain antigens are known to promote Th1-like shifts. Given the viral burden in OSP2 exosomes and that Th1 cells play a central role in protection against viral infections and enhance IFN-γ production to induce early inflammatory responses [54], an increase in Th1-like cells is not unexpected. Interestingly, exosome-stimulated DC-mediated T-cell responses can also promote Th2-like differentiation via OX40L [55].

The subtypes of HTLV have strikingly different clinical outcomes. HTLV-1 is conclusively associated with several pathologies, including ATLL and HAM/TSP, while HTLV-2 is often associated with elevated lymphocytes and an overall increase in cancer mortality. Contrary to the differences in homology (~30%), they share considerable resemblances in genome structure; replication pattern; and structural, regulatory, and accessory protein properties. However, HTLV-1 is tropic towards CD4+ T cells, while HTLV-2 favors CD8+ T cells in vivo [42]. Additionally, the viral protein HBZ is unique to HTLV-1 while the other subtypes express its counterpart, APH-2 [56]. Our data are the first to illustrate the differences in exosome analyte cargo as well as cytokines regarding HTLV types. The lack of difference between analytes in ACs and HAM/TSP patients highlights the limited pathology of HTLV-2. Indeed, further studies are essential to profile distinct differences in exosome cargo and how it may influence overall HTLV-2-linked pathology.

Exosomes carrying ICPs impact immune cell function by suppressing activation or maturation, which can be restored by blocking exo-ICPs, with the mAb cocktail targeting select ICPs. Although underlying mechanisms remain elusive, understanding the potential these exosomes in influencing T-cell function can provide insights into therapeutic strategies. Interestingly, we previously demonstrated that healthy T cells exposed to OSP2-derived exosomes decrease TNF-α and MIP-1α expression, while IFN-γ and IL-2 remained unchanged [35]. TNF-α is elevated in the cerebrospinal fluid of HAM/TSP patients and released by infiltrating infected perivascular lymphocytes, which are also higher in HAM/TSP patients [57]. MIP-1α, a crucial chemokine in initiating inflammation, is inversely correlated with proviral load in HAM/TSP patients [8]. Furthermore, the production of granzyme B and perforin is a hallmark of cytolytic potential, and while naïve T cells are activated by exosomes alone, our data suggest that, following exosome exposure, the overall T-cell activation is dampened, while targeting ICPs can recover T-cell activity. This dampening effect of activated cells is demonstrated by a reduction in phosphorylated Akt/ERK, which is upstream from NFκB, mTORC, and CREB, promoting different immune responses in both CD4+ and CD8+ T cells [58]. Overall, our study suggests a novel role that infected cell-derived exosomes have in perpetuating dysfunctional T cells, a hallmark of HAM/TSP pathogenesis, and offers insights into potential therapeutic strategies.

## Figures and Tables

**Figure 1 viruses-16-01443-f001:**
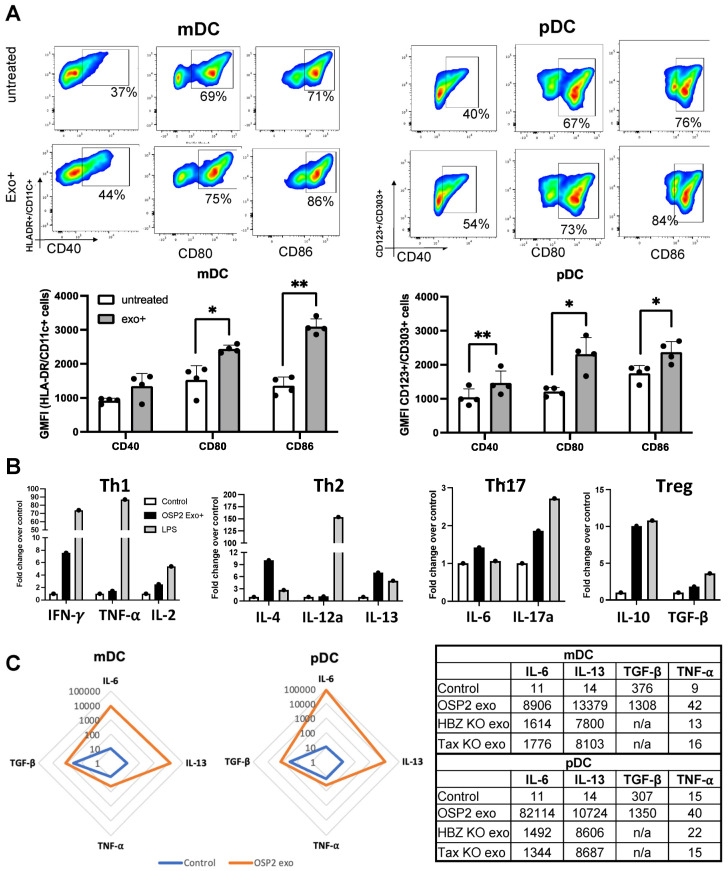
Exosomes activate and elicit proinflammatory responses in dendritic cells. (**A**) Representative phenotype of myeloid dendritic cell (mDC) and plasmacytoid dendritic cell (pDC) population differences with exosome stimulation (*n* = 4). GMFI levels of CD40, CD80, and CD86 in mDC and pDC populations untreated or stimulated with exosomes. Bars represent mean ± standard deviation (SD). (**B**) Expression of cytokines associated with Th1, Th2, Th17, and Treg polarization in total DCs stimulated with OSP2 cell-derived exosomes or LPS. Cytokines were grouped according to associated Th1 (IFN-γ, TNF-α, and IL-2), Th2 (IL-4, IL-12a, and IL-13), Th17 (IL-6 and IL-17a), and Treg (IL-10 and TGF-β) subsets. (**C**) Cytokine levels (pg/mL) in supernatants of mDCs (**left**) and pDCs (**right**) untreated or exosome-stimulated. Statistical differences were determined by one-way ANOVA, * *p* < 0.05, ** *p* < 0.005.

**Figure 2 viruses-16-01443-f002:**
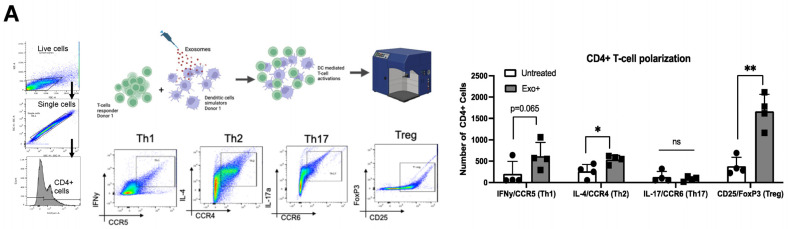
Dendritic cell-pulsed exosomes polarize T cells. (**A**) Exosome stimulation schematic and representative gating strategy. Donor CD3+ T cells and matched total DCs were isolated (*n* = 4). (**B**) DCs were exposed to OSP2-derived exosomes for 24 h and subsequently co-incubated with donor-matched T cells for another 24 h. Absolute count of CD4+ T-cells expressing subtype-associated markers: Th1 (IFN-γ, CCR5), Th2 (IL-4, CCR4), Th17 (IL-17, CCR6), and Treg (CD25, FoxP3). Quantification (pg/mL) of IFN-γ, TGF-β, and TNF-α cytokine levels in exosome-stimulated DC–T-cell co-culture (*n* = 3). (**C**) Representative flow plots and quantification (*n* = 4) of percent positive CD4+ T cells expressing functional markers of IFN-γ, IL-4, IL-17a, CD25, CCR5, and CCR6 after stimulation with OSP2 cell-derived exosomes. Bars represent mean ± standard deviation (SD). Statistical differences were determined by one-way ANOVA, * *p* < 0.05, ** *p* < 0.005, *** *p* < 0.001.

**Figure 3 viruses-16-01443-f003:**
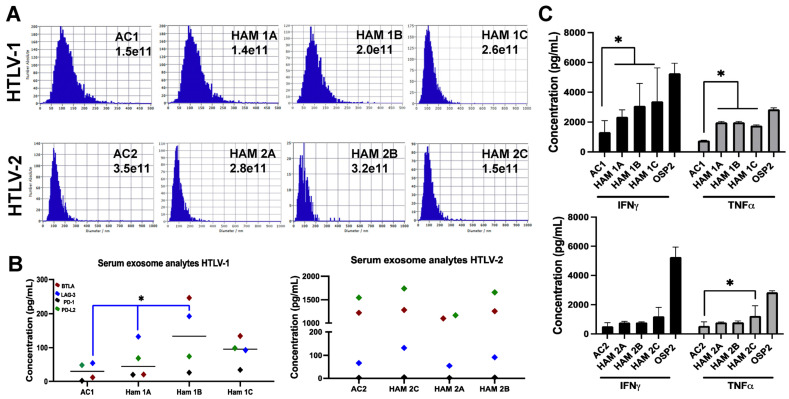
Immune checkpoint and anti-viral cytokine expression of exosomes from individuals with HAM/TSP. (**A**) Representative NTA of exosomes isolated from asymptomatic carrier (AC, *n* = 1) and HAM/TSP (HAM, *n* = 3) patient sera in individuals infected with HTLV-1 or HTLV-2. (**B**) Quantification of immune checkpoint proteins BTLA, LAG-3, PD-1, and PD-L2 in exosomes from AC and HAM/TSP patient sera from individuals infected with HTLV-1, left, or HTLV-2, right. (**C**) Quantification of IFN-γ and TNF-α levels in exosomes from AC and HAM/TSP patient sera from individuals infected with HTLV-1, left, or HTLV-2, right. Bars represent mean ± standard deviation (SD). Statistical differences were determined by unpaired two-tailed T-tests of technical replicates, * *p* < 0.05.

**Figure 4 viruses-16-01443-f004:**
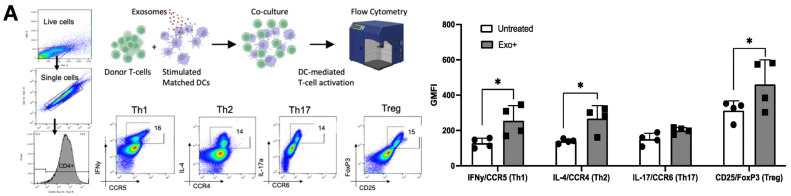
HAM/TSP exosomes skew Th1/Treg responses and sensitize cells toward infection. (**A**) Exosome stimulation schematic and representative gating strategy. CD3+ T cells and total DCs were isolated from matched donors (*n* = 4). DCs were stimulated with exosomes from HTLV-1 patient sera for 24 h and subsequently co-incubated with donor-matched T cells for another 24 h. Differences in GMFI of representative markers are plotted. Bottom, absolute count of CD4+ T cells expressing subtype-associated markers: Th1 (IFN-y and CCR5), Th2 (IL-4 and CCR4), Th17 (IL-17 and CCR6), and Treg (CD25 and FoxP3). (**B**) Quantification (pg/mL) of IFN-γ, TGF-β, and TNF-α cytokine levels in patient exosome-stimulated DC–T-cell co-culture. (**C**) Representative flow plots and quantification (*n* = 4) of percent positive CD4+ T cells expressing functional markers of IFN-γ, IL-4, IL-17a, CD25, CCR5, and CCR6 after stimulation with HTLV-1 patient-derived exosomes. Bars represent mean ± standard deviation (SD). Statistical differences were determined by one-way ANOVA, * *p* < 0.05, ** *p* < 0.005, *** *p* < 0.001.

**Figure 5 viruses-16-01443-f005:**
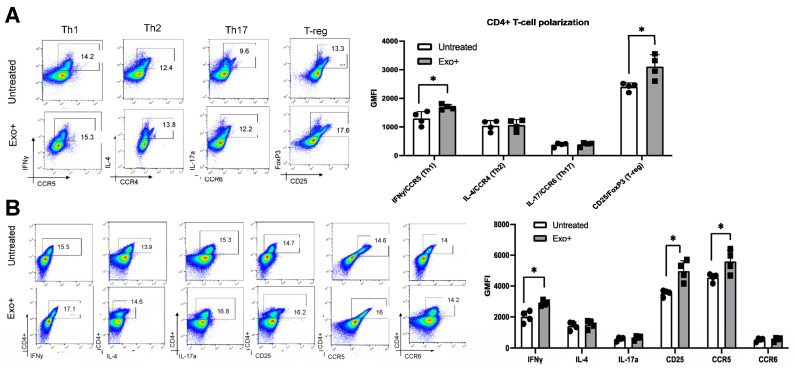
HAM/TSP exosomes skew T cells to Th1/Treg profiles in patient PBMCs. (**A**) Representative flow plots and quantification (*n* = 4) of the polarization of T cells from individuals with HTLV-1 with or without HTLV-1 sera-derived exosomes. (**B**) Representative flow plots and quantification (*n* = 4) of HTLV-1-infected patient CD4+ T cells with or without HTLV-1 sera-derived exosome stimulation. Bars represent mean ± standard deviation (SD). Statistical differences were determined by one-way ANOVA, * *p* < 0.05.

**Figure 6 viruses-16-01443-f006:**
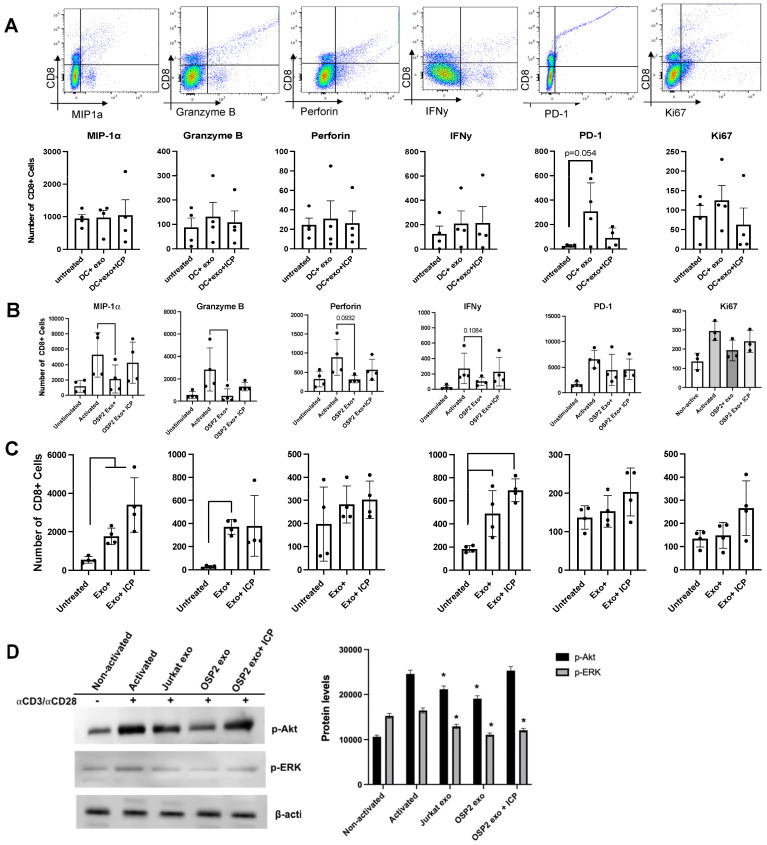
Exosomes diminish CD8+ T-cell activity. (**A**) Schematic of treatment and gating strategy of CD8+ T cells. CD3+ T cells and total DCs were isolated from matched donors (*n* = 4). DCs were stimulated with OSP2-derived exosomes for 24 h and subsequently co-incubated with donor-matched T cells for another 24 h. Absolute counts of CD8+ T cells expressing MIP-1a, Granzyme B, Perforin, IFN-γ, PD-1, and Ki67 after co-incubation with exosome-pulsed DCs. (**B**) Absolute counts of CD8+ T cells after CD3+/CD28+ activation and stimulation with OSP2 exosomes alone, or exosomes incubated with a cocktail of anti-PD-L2, anti-BTLA, anti-LAG-3, and anti-PD-1 blocking antibodies. (**C**) Absolute counts of CD8+ T cells expressing MIP-1a, Granzyme B, Perforin, IFN-γ, PD-1, and Ki67 in HTLV-1-infected patient T cells (*n* = 4) stimulated with patient sera exosomes alone or exosomes incubated with a cocktail of anti-PD-L2, anti-BTLA, anti-LAG-3, and anti-PD-1 blocking antibodies. Counts were determined by experiments with *n* = 3/4 healthy donors, and error bars represent SEM of donor variation. Bars represent mean ± standard deviation (SD). Statistical differences were determined by one-way ANOVA. (**D**) Western blot and densitometry of CD3/CD28-stimulated T cells treated with Jurkat and OSP2 exosomes alone, or exosomes incubated with a cocktail of anti-PD-L2, anti-BTLA, anti-LAG-3, and anti-PD-1 blocking antibodies. Blot was probed for phosphorylated AKT or ERK. Error bars represent SEM of blot variation; statistical differences were determined by paired two-tailed T-test, * *p* < 0.05.

## Data Availability

All data generated or analyzed during this study are included in this published article (and its Appendix A).

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
