# Peer review of "Dendritic Cells Pulsed with HAM/TSP Exosomes Sensitize CD4 T Cells to Enhance HTLV-1 Infection, Induce Helper T-Cell Polarization, and Decrease Cytotoxic T-Cell Response"

_viruses, 2024, doi:10.3390/v16091443_

Round 1

Reviewer 1 Report

Comments and Suggestions for Authors

The manuscript is well-documented and presented in an excellent way. It has the potential to be a valuable reference for other manuscripts in this field and provide new insights.

I have only a few minor comments:

1. It is necessary to specify the number of participants.

2. What was the purity of the magnetically isolated populations?

3. Generally, the figures are well presented. I have a single remark about Figure 4 (Panel A, „Exosome simulation schematic…") the description in panel A is illegible.

4. The same, Supplemental Figure 1. Exosome-mediated activation of DC subsets. A) Gating strategy for pDC and mDC populations...  On the first three dot plots, gates description is illegible.

5. Is the mean or median displayed in the graphs?

Reviewer 2 Report

Comments and Suggestions for Authors

The manuscript presents novel results obtained by analyzing the impact of exosomes derived from HTLV-1-infected cells and sera from patients affected by HTLV-1-associated myelopathy/tropical spastic paraparesis (HAM/TSP) on dendritic cells and T-cells. The results show that exosomes in the context of HAM/TSP alter T-cell activity with possible effects on HAM/TSP pathogenesis. The methodology applied is rigorous, as are the controls and techniques used. The figures efficiently represent the results in detail. The discussion is thorough and includes a good comparison with the literature. Original data on the content of HTLV exosomes may be of reliable interest in research on HTLV infection and associated diseases.

Reviewer 3 Report

Comments and Suggestions for Authors

The authors reported dendritic cells pulsed with HAM/TSP exosomes (Exo) sensitize CD4 T cells to enhance HTLV-1 infection, induce helper T-cell polarization, and decrease cytotoxic T-cell response.

This reviewer thinks that there are some concerns about the method.

Major points:

About the Materials and Methods (M&M),

1. Do OSP2 (HAM/TSP-derived) cells produce not only exosomes but also HTLV-1 viral particles? If so, there is a possible the fraction of exosome isolated by ultracentrifuge method include HTLV-1 virion. An explanation is required regarding this possibility. In addition, to rule out the possibility of contaminated HTLV-1 virion into exosome fraction, it is necessary to clarify HTLV-1 virion proteins such as gp46 by immunoblotting using the fraction of exosome.

2. Please describe the volume of peripheral blood obtained from HTLV-1-infected individuals for isolation of DCs in the M&M section.

3. Please describe in detail the method to quantify proinflammatory cytokine levels of exosomes. Although authors used ELISAs, is there any pretreatment for extracting or eluting proinflammatory cytokines contained in exosomes? Please explain the principle by which cytokines contained in exosomes bind to antibodies in ELISA.

Round 2

Reviewer 3 Report

Comments and Suggestions for Authors

I have read the revised manuscript, which has been appropriately revised according to reviewers recommendations.